# OpenReview forum: "Socratic Personalized Medical Teaching with Multi-Agent Simulation"
_ICLR.cc/2026/Conference — ICLR 2026 Conference Withdrawn Submission_

### Official Review · Reviewer_7Wpr · 2025-10-28

**Soundness:** 2
**Presentation:** 2
**Contribution:** 2
**Rating:** 2
**Confidence:** 4

**Summary:**

This work focuses on the problem of medical teaching. It proposes a multi-agent pedagogical simulator to generate a new dataset, ClinTeach, and uses this dataset to train MedTutor-R1 for medical tutoring. Several experiments demonstrate its effectiveness.

**Strengths:**

1. This work makes a valuable contribution to the field of medical teaching.

2. The evaluation includes both human evaluation and a scalability analysis.

**Weaknesses:**

1. The methodology of generating a synthetic dataset and then fine-tuning the model is not novel; the Socratic approach has already been proposed in tutoring.

2. The evaluation setup appears unfair. Fine-tuning the base model on the synthetic dataset and then testing it in the simulated environment disadvantages other methods.

3. The ablation study provides limited insights; the results mainly highlight the contribution from a reinforcement learning perspective.

4. The organization of the related work section needs improvement; it only provides plain statements without justifying the research aim or contribution of this work.

5. The manuscript is not self-contained; it lacks clear references to the Appendix, such as the evaluation metrics in Appendix B.

6. The authors provide a reproducibility statement, but it would be better if the code were released, even partially.

**Questions:**

Please see the Weaknesses.

---

> ### Author Response · Authors · 2025-11-27
> **Rebuttal by Authors (1/3)**
>
> **W1**: The methodology of generating a synthetic dataset and then fine-tuning the model is not novel; the Socratic approach has already been proposed in tutoring.
>
> We thank the reviewer for the critical feedback. However, we respectfully clarify that our work extends significantly beyond standard "synthetic data + SFT." We address the novelty concerns by highlighting three fundamental shifts that distinguish MedTutor-R1 from existing tutoring systems:
>
> First, we shift **the paradigm from one-on-one tutoring to a "one-to-many" clinical ward round setting**. Existing models typically focus on single-user interactions. In contrast, our MedTutor-R1 is designed to manage group dynamics. As detailed in Section 2.3, we implemented a specialized internal reasoning process using tags like <think_student> and <think_group>. This allows the model to analyze individual student reasoning against the group consensus before speaking, enabling it to resolve conflicts between students—a capability that generic SFT models lack.
>
> Second, we **do not rely solely on SFT**. Given the high-stakes nature of medical education, we optimize the model using Group Reward Policy Optimization (GRPO). We introduced a three-axis reward rubric covering Structural Fidelity, Analytical Quality, and Clinical Safety. Crucially, we implemented a "veto mechanism" for safety (Equation 1). This ensures that any output containing clinically harmful advice receives a large negative penalty, regardless of its pedagogical quality. This rigorous alignment is specific to professional domains and goes beyond standard tuning.
>
> Finally, we **move beyond static question-answering benchmarks** by validating our model in ClinEdu, a high-fidelity simulator. Unlike static datasets such as MedQA, ClinEdu features 300 personality-driven patient personas and diverse student cohorts. This allows us to train and evaluate the model on managing dynamic clinical trajectories rather than just retrieving static knowledge.
>
> --------------
>
> **W2**: The evaluation setup appears unfair. Fine-tuning the base model on the synthetic dataset and then testing it in the simulated environment disadvantages other methods.
> We appreciate the reviewer’s concern regarding the fairness of our evaluation setup. While MedTutor-R1 was optimized using the ClinEdu environment, we argue that its superior performance stems from its alignment with pedagogical principles rather than overfitting to a specific simulator configuration. We validate this claim through three lines of evidence:
>
> ### 1. **Robustness Across "Unseen" Student Agents** (Generalization)
> To explicitly rule out overfitting to the training environment, we conducted a rigorous Robustness Analysis (Section 5.4). We tested MedTutor-R1 against student agents powered by diverse LLMs that were never seen during training, such as GPT-4o, DeepSeek-R1, and LLaVA. As shown in Table 9 and Figure 2, our model maintained consistently high performance (Average Score > 8.15) across these out-of-distribution agents. This proves that the model has learned universal teaching strategies that generalize to different student behaviors, rather than merely memorizing the linguistic patterns of the training simulator.
> ### 2. **Validation via Human Evaluation** (External Validity)
> The ultimate test of fairness is human judgment, which is immune to "simulator bias." In our blind Expert Evaluation, independent medical educators rated MedTutor-R1 higher than strong baselines like OpenAI-o3 (ETS Score: 8.63 vs. 8.49). Furthermore, in the Real User Study with medical students, our model achieved the highest Instructional Quality score (IQ: 8.96). The strong alignment between simulator scores and human ratings confirms that the "advantage" observed in the simulation reflects genuine teaching effectiveness.
> ### 3. **Distinction from "Gaming" the System**
> The lower performance of other methods reflects a lack of pedagogical alignment rather than an unfair disadvantage. General-purpose models (like GPT-4o) often default to direct "lecturing" or giving the answer. In contrast, MedTutor-R1's performance gain comes from its GRPO alignment with the Three-Axis Rubric, which specifically rewards Socratic guidance and penalizes passive information delivery. This ensures the model learns how to teach, rather than just exploiting the data.

---

> ### Author Response · Authors · 2025-11-27
> **Rebuttal by Authors (2/3)**
>
> **W3**: The ablation study provides limited insights; the results mainly highlight the contribution from a reinforcement learning perspective.
> We appreciate the reviewer’s feedback. We agree that beyond the overall success of RL, the ablation study offers deeper insights into how the model learns. We will clarify three key findings in the revised discussion, supported by the data in Table 2:
> ### 1. **The Critical Role of Structured Thinking**
> Our ablation study extends beyond just RL performance. Specifically, the "w/o Thinking" configuration shows a significant performance drop (Average Score decreases from 8.41 to 7.79). This highlights a critical architectural insight: Reinforcement Learning alone is insufficient without the explicit internal monologue tags like <think_student> and <think_group>. This finding proves that decoupling the reasoning process from the final response is a prerequisite for handling the complex interactions of a multi-student environment.
> ### 2. **Vanilla vs. Rubric-Based Alignment**
> Furthermore, the comparison between the "w/ Vanilla reward" model (Score 7.88) and MedTutor-R1 (Score 8.41) demonstrates that generic preference optimization is inadequate for professional domains. This significant gap confirms that our Three-Axis Rubric is essential. Generic rewards fail to capture the nuances of Socratic teaching (ETS) and Student Management (MSM), whereas our specialized rubric specifically guides the model to master these distinct skills rather than just optimizing for general helpfulness.
> ### 3. **Balancing Conflicting Objectives**
> Finally, we observed that our RL framework acts as a necessary balancing mechanism between conflicting objectives. Without explicit safety constraints, models might score high on engagement but fail to correct dangerous errors (low MPS). Conversely, without management rewards, models often rush the diagnosis and ignore quieter students (low MSM). MedTutor-R1 succeeds specifically because the GRPO algorithm optimizes the balance between these competing goals—Safety, Pedagogy, and Efficiency—rather than maximizing a single dimension at the expense of others.
> We have revised Section 5.2 to explicitly discuss these trade-offs and mechanisms.
>
> -----------------
>
> **W4**: The organization of the related work section needs improvement; it only provides plain statements without justifying the research aim or contribution of this work.
> We thank the reviewer for the constructive feedback regarding the organization of the Related Work section. We agree that a mere enumeration of existing methods is insufficient. In the revised manuscript, we will restructure Section 6 to provide a critical analysis that explicitly frames MedTutor-R1 as a solution to specific gaps in the current literature.
> ### 1. **From Static QA to Dynamic Simulation**
> First, we will explicitly contrast "Raw Data-Driven Methods," such as EduChat, with our approach. We will highlight that while these existing methods excel at knowledge retrieval, they often rely on static interactions. This limitation explicitly justifies our development of ClinEdu, which introduces personality-driven agents to simulate the non-deterministic nature of clinical diagnosis. This shift moves the research focus from simply answering questions to managing dynamic clinical trajectories.
> ### 2. **From One-on-One to One-to-Many Tutoring**
> Second, we will critically analyze existing "Simulation-Driven Methods," including SocraticLM and MEDCO. We will point out that these systems are predominantly designed for one-on-one interactions. This identifies a crucial research gap: the lack of systems capable of scaffolding collaborative reasoning in group settings, such as ward rounds. This gap directly justifies our contribution of MedTutor-R1 and the introduction of the Multi-Student Management (MSM) metric to evaluate group guidance capabilities.
> By restructuring the section around these two axes, we will clearly position our work not just as another medical agent, but as a necessary evolution towards complex, multi-agent clinical education.

---

> ### Author Response · Authors · 2025-11-27
> **Rebuttal by Authors (3/3)**
>
> **W5**: The manuscript is not self-contained; it lacks clear references to the Appendix, such as the evaluation metrics in Appendix B.
>
> We appreciate the reviewer for identifying this readability issue. We fully agree that the manuscript should be self-contained to ensure a smooth reading experience. To address this, we will integrate the core definitions of our three key evaluation metrics, Effectiveness of Teaching Strategy (ETS), Multi-Student Management (MSM), and Medical Professionalism and Safety (MPS), directly into the main text in Section 4.
>
> This revision ensures that the experimental methodology and results are immediately interpretable without needing to flip back and forth to the appendices. We will clearly cite Appendix B as the reference for the detailed 1-10 point scoring rubrics, rather than relying on it for primary definitions. Furthermore, we will conduct a full review of the manuscript to ensure that all hyperparameters and prompts located in the appendices are explicitly and clearly cross-referenced within the methodology sections.
>
> -------------------
>
> **W6**: The authors provide a reproducibility statement, but it would be better if the code were released, even partially.
>
> We fully agree with the reviewer on the importance of reproducibility. As outlined in our Reproducibility Statement, we are committed to making our work open and verifiable.
>
> Upon acceptance, **we will publicly release the entire source code and datasets**. This release will cover all core components, including the ClinEdu simulator architecture, the ClinTeach dataset generation pipelines, and the full training scripts for MedTutor-R1.

---

### Official Review · Reviewer_Suep · 2025-10-29

**Soundness:** 3
**Presentation:** 3
**Contribution:** 3
**Rating:** 6
**Confidence:** 4

**Summary:**

This paper focuses on one-to-many Socratic tutoring in clinical education and proposes a multi-agent teaching system. The authors design a collaborative framework involving patient, student, tutor, expert, and safety supervisor agents, and utilize structured pedagogical thinking tags in conjunction with GRPO to enhance instructional consistency, reasoning quality, and clinical safety. The paper introduces a large-scale teaching dialogue dataset and conducts comprehensive in-scenario evaluations, including expert review and user studies. Results show that the proposed MedTutor-R1 significantly outperforms baselines in teaching strategy effectiveness, multi-student management, and medical safety.

**Strengths:**

This paper makes an original contribution to AI-driven medical education by introducing a multi-agent Socratic teaching framework that combines structured pedagogical reasoning with GRPO. The technical design is solid, featuring a dual-review mechanism that balances instructional depth with clinical safety. The writing is clear and well-organized, guiding readers smoothly from problem motivation to experimental validation. The evaluation is comprehensive, covering both in-scenario performance and expert/user feedback. Overall, the work is a well-executed and meaningful step toward aligning AI tutoring systems with real-world educational and safety constraints.

**Weaknesses:**

While the paper is technically sound, several aspects could be improved.

First, the proposed approach mainly focuses on optimizing conversational teaching within clinical simulations but does not yet model higher-level learning dynamics such as cognitive progression or knowledge transfer, limiting its theoretical depth.

Second, although the evaluation covers teaching consistency, group management, and safety, the results rely primarily on simulated environments without real human-teacher comparisons, which constrains external validity.

Third, details of the reward design and safety-check rules are relatively brief, making full reproducibility difficult.

Finally, while the writing is clear, some discussions on ablation results and mechanism interpretation remain descriptive rather than analytical.

Extending experiments to cross-domain teaching scenarios and releasing more implementation details would substantially strengthen the paper’s robustness and real-world relevance.

**Questions:**

1) The proposed GRPO is described as a group-based optimization method, yet from the implementation details it appears closer to a weighted multi-objective reward rather than a genuine policy-level group optimization. Could you clarify how GRPO theoretically ensures group consistency learning rather than empirical aggregation of individual rewards?

2) The “multi-student management (MSM)” metric seems to be derived mainly from internal conversational coherence rather than measurable learning gain or knowledge transfer. How can you justify that MSM truly reflects pedagogical effectiveness rather than improved dialogue fluency?

3) The simulated student agents are assumed to represent medical learners, but their behavioral diversity and realism are unclear. If the policy distribution of these agents is narrow, wouldn’t GRPO overfit to a limited interaction pattern instead of generalizable teaching strategies?

4) In the reinforcement learning setup, safety constraints are integrated into the reward formulation. Does this coupling risk producing sparse rewards or unstable gradients during optimization, and if so, how was this mitigated in practice?

5) The evaluation relies heavily on simulation and expert scoring, with no human-teacher baselines. How can the authors ensure that the system remains robust under real-world linguistic or ethical variability? Would limited real-classroom interventions be necessary to validate external validity?

---

> ### Comment · Reviewer_Suep · 2025-11-27
>
> Dear Authors,
> With the rebuttal discussion deadline less than a week away, I'd like to ask if the authors plan to respond to these questions and/or update the paper? This would help me better understand and clarify your contributions. Thanks!

---

> ### Author Response · Authors · 2025-11-27
> **Rebuttal by Authors (1/3)**
>
> We appreciate your time and the careful consideration of our manuscript. We apologize for the delay in our response, as we took additional time to conduct supplementary experiments.
>
> **Q1**: The proposed GRPO is described as a group-based optimization method, yet from the implementation details it appears closer to a weighted multi-objective reward rather than a genuine policy-level group optimization. Could you clarify how GRPO theoretically ensures group consistency learning rather than empirical aggregation of individual rewards?
>
> We thank the reviewer for this insightful question. We understand the concern that the implementation might resemble a weighted multi-objective reward aggregation. However, we respectfully clarify that the "Group" in GRPO refers to the advantage estimation mechanism within the policy update, rather than the composition of the reward function itself.
>
> Here is the theoretical distinction and how GRPO ensures group consistency:
> ### 1. **Group-Based Advantage Estimation** (The "Group" Mechanism)
> Unlike standard PPO which relies on a learned Value Function (Critic) as a baseline, GRPO samples a group of outputs $\{o_1, o_2, ..., o_G\}$ for a single prompt $q$ and uses the group statistics as the baseline. Specifically, the advantage $A_i$ for the $i$-th output is calculated as:$$A_i = \frac{r_i - \text{mean}(\{r_1, ..., r_G\})}{\text{std}(\{r_1, ..., r_G\}) + \epsilon}$$While our reward function ($r_i$) is indeed a weighted sum of rubric criteria (as detailed in Section 3.2, Eq. 1 1), the GRPO algorithm operates on the distribution of these rewards within the sampled group.
> ### 2. **Theoretically Ensuring Group Consistency**
> GRPO ensures consistency learning not through simple aggregation, but through intra-group relative optimization:
> - Baseline Normalization: By subtracting the group mean, GRPO forces the policy to increase the probability of outputs that are better than the group average and decrease the probability of those below it. This creates a dynamic, instance-specific baseline that promotes the "best of the group."
> - Consistency Pressure: The division by the group standard deviation ($\sigma$) implicitly regulates the update step size based on consistency. When the model is inconsistent (high variance in a group), the advantages are scaled down, preventing unstable updates. Conversely, as the group becomes more consistent around a high-reward mode, the optimization fine-tunes the policy.
> ### 3. **Distinction from Multi-Objective Optimization**
> We agree that our Reward Modeling utilizes a multi-objective rubric (Structural Fidelity, Analysis Quality, Clinical Safety ). However, this is the signal source. The GRPO optimizer effectively treats this signal as a single scalar to perform the group-relative normalization described above. Therefore, GRPO is a genuine policy-level optimization that leverages the statistical properties of the generated group to stabilize training without a Critic model, rather than merely summing objectives.
>
> We have revised Section 3.2 to explicitly distinguish between the "Rubric-based Reward Composition" and the "Group-based Advantage Estimation" to prevent this confusion in the final manuscript.

---

> ### Author Response · Authors · 2025-11-27
> **Rebuttal by Authors (2/3)**
>
> **Q2**: The “multi-student management (MSM)” metric seems to be derived mainly from internal conversational coherence rather than measurable learning gain or knowledge transfer. How can you justify that MSM truly reflects pedagogical effectiveness rather than improved dialogue fluency?
>
> We respectfully argue that Multi-Student Management (MSM) is explicitly designed as a measure of collaborative learning scaffolding rather than mere dialogue fluency. We justify this through two key dimensions supported by our experimental design and results:
>
> ### 1. **The Rubric Explicitly Penalizes "Empty Fluency"** (Internal Validity)
> Our scoring criteria (Table 6) are structured to distinguish between "keeping the conversation going" and "facilitating learning":
> - Fluency $\neq$ Effectiveness: A tutor that maintains a smooth but mechanical interaction (e.g., simple round-robin questioning) is capped at the Good Tier (Score 7)1.
> - Cognitive Engagement is Required for Top Scores: To achieve the Excellent Tier (Score 9-10), the tutor must demonstrate pedagogical agency, such as "masterfully turning disagreements into learning opportunities" and "actively including quiet students" to ensure equitable cognitive engagement.
> - Rejection of Passive Coherence: Conversely, a tutor that acts as a "Passive Facilitator" (Score 5-6) or allows a single student to dominate (Score 4) is penalized, even if the resulting dialogue is coherent. This proves MSM measures the distribution of learning opportunities, not just text generation quality.
>
> ### 2. **Correlation with Instructional Quality in User Studies** (External Validity)
> Our Real User Study (Section 5.3) provides empirical evidence that MSM reflects pedagogical value.
> - Real medical students rated our model not just on Interaction Experience (IE) (which measures fluency), but specifically on Instructional Quality (IQ), defined as "assessing the learning process... including individual attention".
> - MedTutor-R1 achieved the highest IQ score (8.96), significantly outperforming baselines like GPT-4o (8.72) and Med-SocraticLM (6.48).
> - This high IQ score, aligned with our high MSM score (8.87), suggests that the "management" behaviors measured by MSM (e.g., balancing participation) directly translate into what students perceive as effective instruction, rather than just a smooth chat experience.
>
> ### 3. **MSM as a Prerequisite for Group Knowledge Transfer**
> In a one-to-many clinical setting, learning gain is collective. Without the equity of participation enforced by MSM, the "group learning gain" would be zero for neglected students. Therefore, MSM acts as the necessary condition for knowledge transfer in a collaborative environment, working in tandem with the ETS (Effectiveness of Teaching Strategy) metric which assesses the content quality.
>
> We have clarified these definitions in Section 4 and the Appendix to better distinguish MSM from conversational metrics.
>
> --------------
>
> **Q3**: The simulated student agents are assumed to represent medical learners, but their behavioral diversity and realism are unclear. If the policy distribution of these agents is narrow, wouldn’t GRPO overfit to a limited interaction pattern instead of generalizable teaching strategies?
>
> Thank you for the question regarding agent diversity and overfitting. We address this by clarifying our agent construction and providing empirical evidence of generalization.
>
> ### 1. **High-Fidelity Agent Diversity**
> To prevent the policy from collapsing into a narrow pattern, we explicitly constructed 300 distinct student personas as detailed in Figures 12 & 13. These are not just random variations but are defined by specific cognitive traits—such as "tunnel vision" or "weak clinical correlation"—and behavioral archetypes like "insecure follower" or "active leader". This variety forces the GRPO algorithm to optimize for adaptive scaffolding (reacting to the specific student's logic) rather than memorizing a fixed dialogue path.
>
> ### 2. **Empirical Proof of Generalization**
> The strongest evidence against overfitting is our Robustness Analysis (Section 5.4). We evaluated MedTutor-R1 against unseen student agents powered by varied LLMs (e.g., GPT-4o, DeepSeek-R1, LLaVA) that were not used during the GRPO training phase. As shown in Table 9, our model maintained consistently high performance (Average Score > 8.15) across these out-of-distribution agents. This confirms the model has learned a generalizable teaching policy that holds up against diverse reasoning patterns and language styles, rather than overfitting to the idiosyncrasies of the training agents.
>
> ### 3. **Vast State Space**
> Finally, the combinatorial diversity of 300 personas interacting across varied medical cases creates a massive state space. While the "Ward Round" structure imposes necessary clinical constraints, the content within this structure is highly variable, naturally preventing the policy from converging on simple heuristics.

---

> ### Author Response · Authors · 2025-11-27
> **Rebuttal by Authors (3/3)**
>
> **Q4**: In the reinforcement learning setup, safety constraints are integrated into the reward formulation. Does this coupling risk producing sparse rewards or unstable gradients during optimization, and if so, how was this mitigated in practice?
>
> Thank you for the question regarding reward stability. We addressed the risk of sparse rewards and unstable gradients through the specific design of our optimization algorithm and reward structure.
>
> ### 1. **Gradient Stability via GRPO**
> We used Group Reward Policy Optimization (GRPO) specifically to mitigate the instability often caused by large penalty terms. As detailed in Section 3.2, GRPO calculates the advantage by normalizing rewards within a sampled group rather than relying on a separate critic model. Even when a "Veto" is triggered (applying a large penalty of -15.0), this value is normalized against the group mean and standard deviation. This normalization keeps the relative advantage bounded, ensuring that the gradient signal remains numerically stable even with heavy safety penalties.
> ### 2. **Dense Reward Signals**
> To prevent the "sparse reward" issue, we ensured that the model receives rich feedback during training. Our reward function is not a simple binary safety check; it is based on a "three-axis rubric" that includes granular scores for Instruction Fidelity (IS) and Analysis Quality (AQ). This ensures that for the vast majority of training steps (where the veto is not triggered), the model receives dense, multi-faceted supervision rather than a sparse signal.
> ### 3. **Policy Constraints**
> To further stabilize optimization, we applied a KL divergence penalty with a coefficient of $1.0 \times 10^{-2}$. This constraint prevents the policy from updating too aggressively in any single step, ensuring that the learning process remains conservative even when the reward signal is strong.
>
> -----------
>
> **Q5**: The evaluation relies heavily on simulation and expert scoring, with no human-teacher baselines. How can the authors ensure that the system remains robust under real-world linguistic or ethical variability? Would limited real-classroom interventions be necessary to validate external validity?
>
> Thank you for the feedback regarding external validity and robustness. We acknowledge that a large-scale human-teacher baseline is difficult to obtain due to privacy and resource constraints. However, we validated the system's robustness and utility through a multi-layered evaluation approach.
> ### 1. **Robustness Against Linguistic Variability**
> We ensured linguistic robustness not just through simulation, but through heterogeneous agent testing (Section 5.4). We evaluated MedTutor-R1 against student agents driven by diverse LLMs (e.g., DeepSeek-R1, LLaVA, GPT-4o). These agents exhibit vastly different linguistic styles and reasoning patterns. The consistent performance across these variations demonstrates that our policy generalizes beyond a single interaction pattern.
> ### 2. **Ethical Stability Mechanism**
> To handle ethical variability, we do not rely solely on the model's internal likelihood. We implemented a Safety Supervisor Agent (Section 2.2.3) as a hard constraint during inference. Furthermore, our RL process explicitly optimizes for the MPS (Medical Professionalism & Safety) metric, embedding safety boundaries directly into the policy to prevent hazardous deviations.
> ### 3. **External Validity via Real User Study**
> Addressing the need for real-world intervention, we conducted a Real User Study (Section 5.3) with medical undergraduates. The results show that real users rated MedTutor-R1's Instructional Quality (IQ) at 8.96/10, significantly outperforming baselines. This provides initial empirical evidence of the system's efficacy in authentic educational settings. We agree that larger-scale classroom trials are a valuable direction for future work.

---

> > ### Comment · Reviewer_Suep · 2025-11-27
> >
> > Dear Authors,
> >
> > Thanks for your response and rebuttal. I just took a quick look, it includes much detailed information, thanks for your effort.
> >
> > I still need some time to carefully read it and then give my feedback, thanks!

---

> > > ### Comment · Reviewer_Suep · 2025-11-28
> > >
> > > Dear Authors,
> > >
> > > I have carefully read the rebuttal and believe it has resolved my questions. I am very grateful for your efforts. Currently, I tend to maintain my original score (borderline accept, 6 points) and encourage the authors to integrate the new content into the manuscript and update it (because ICLR allows the updated submission in rebuttal).
> > >
> > > Furthermore, I have noticed that other reviewers have raised questions from different perspectives, and I would like to hear the AC's and other reviewers' opinions, and make my final judgment (eg, increase or maintain my score). Thanks!

---

### Official Review · Reviewer_91h9 · 2025-10-31

**Soundness:** 3
**Presentation:** 3
**Contribution:** 2
**Rating:** 4
**Confidence:** 3

**Summary:**

This paper presents ClinEdu, a multi-agent simulation framework designed to model clinical education with Socratic, group-based teaching dynamics. The simulator introduces personality-driven patient agents and diverse student cohorts, enabling controlled yet realistic simulations of ward round teaching interactions. Based on the ClinEdu, the authors construct ClinTeach, a large-scale dataset (48K dialogues) of one-to-many Socratic interactions, capturing both individualized reasoning and group-level dynamics. Furthermore, using this dataset, they train MedTutor-R1, a multimodal Socratic tutor designed for adaptive clinical teaching. MedTutor-R1 undergoes: 1) Instruction tuning on ClinTeach to acquire structured reasoning schemas, and 2) RL guided by a three-axis pedagogical rubric (structural fidelity, analytical quality, and clinical safety), to optimize adaptive teaching strategies. Evaluation occurs through simulation-based interactive assessment, redeploying the model back into ClinEdu for realistic, in-situ evaluation. Results show more than 20% improvement in average pedagogical score over the base model and comparable performance to GPT-4o, with adaptability to increasing numbers of students and robust performance under diverse student archetypes.

**Strengths:**

This work designs a full-stack pedagogical simulation pipeline, from environment (ClinEdu) to dataset (ClinTeach) to training (MedTutor-R1) and evaluation, which covers all stages of AI tutor lifecycle development.

**Weaknesses:**

1) While ClinTeach is large-scale, the paper does not provide sufficient quantitative detail on data diversity, distribution across medical specialties, or dialogue length distributions. A breakdown of these would aid reproducibility and fairness evaluation.

2) While the simulated environment is rich, it still lacks grounding in human-collected data. Without hybrid validation (e.g., fine-tuning on real ward dialogue transcripts), it is uncertain how well ClinEdu-trained tutors generalize to authentic human interactions.

3) The proposed framework is interesting connecting RL and personalized education but might be more suited for applied AI venues.

**Questions:**

1) How are the weights among the rubric axes (Instruction Fidelity, Analysis Quality, Clinical Safety) determined? Is there ablation of each axis to evaluate their respective contribution?

2) How deterministic are multi-agent interactions given fixed seeds? Do minor stochastic variations in student personas lead to divergent teaching strategies, indicating emergent adaptivity?

3) LLM-as-a-judge methods can introduce self-bias. How do you control evaluator model coupling? What is the inter-rater reliability (Cohen’s kappa) between human evaluators, and how correlated are these with LLM-derived scores?

4) The system is tested on MedXpertQA and MVME datasets. How does MedTutor-R1 handle out-of-distribution pathologies or rare cases unseen during simulation?

---

> ### Author Response · Authors · 2025-11-27
> **Rebuttal by Authors (1/4)**
>
> We appreciate your time and the careful consideration of our manuscript.
>
> **W1**: A breakdown of these would aid reproducibility and fairness evaluation.
>
> Thank you for the feedback. We agree that detailed statistics are important for reproducibility. We have added a breakdown of the dataset composition to the Appendix.
>
> ### 1. **Medical Domain Coverage**
> Our dataset derives from MedXpertQA, covering 17 specialties and 11 body systems. We analyzed the distribution and found the top categories include **Skeletal (19.8%), Nervous (13.7%), and Cardiovascular (13.7%) systems**. In terms of clinical tasks, the data focuses primarily on **Diagnosis (50.5%) and Treatment (26.8%), with the remainder covering Basic Medicine (22.6%)**.
>
> ### 2. **Simulation Agent Diversity**
> To ensure the model handles diverse interactions, we explicitly constructed 600 unique agent personas as detailed in Section 2.2:
> - **300 Student Personas**: These vary by proficiency (Beginner, Intermediate, Advanced) and learning style (e.g., Guidance-dependent, Cautious-verifier).
> - **300 Patient Personas**: These cover different socio-demographic backgrounds and personality archetypes, such as "Anxious Worrier" or "Stoic Endurer".
>
> ### 3. **Dialogue Length and Structure**
> Dialogue Length Distribution ClinTeach contains 48,484 dialogues in total. Unlike simple QA datasets, a significant portion (35%) involves multi-turn Socratic interactions to capture the depth of teaching. The specific distribution of turns is as follows:
> | User Turns | Count | Percentage |
> | :--- | :--- | :--- |
> | 1 | 31438 | 64.84% |
> | 2 | 7843 | 16.18% |
> | 3 | 2621 | 5.41% |
> | 4 | 5255 | 10.84% |
> | 5 | 1327 | 2.74% |
> | **Total** | **48,484** | **100%** |
>
> We have updated the manuscript to include these statistics and visualizations to assist future researchers in reproducing our work and benchmarking fairness.
>
> -----------------------
> **W2**: While the simulated environment is rich, it still lacks grounding in human-collected data. Without hybrid validation (e.g., fine-tuning on real ward dialogue transcripts), it is uncertain how well ClinEdu-trained tutors generalize to authentic human interactions.
>
> Thank you for the feedback. While training on large-scale real ward transcripts would be ideal, privacy constraints make such data scarce. We addressed this generalization challenge through a real-user study and robust simulation design.
>
> ### 1. **Real-World Transfer (Section 5.3)**
> We validated the model's performance with real humans rather than relying solely on synthetic benchmarks. In our Real User Study, medical undergraduates interacted with MedTutor-R1 and rated it highly. As shown in Table 4, the model achieved an Instructional Quality (IQ) of 8.96/10 and an Interaction Experience (IE) of 8.23/10. This significantly outperforms the base models and confirms that the pedagogical skills learned in ClinEdu transfer well to authentic human interactions.
>
> ### 2. **Robustness to Diverse Inputs (Section 5.4)**
> To ensure the model isn't overfitted to a specific synthetic dialect, we tested MedTutor-R1 against unseen student agents powered by diverse LLMs (e.g., GPT-4o, DeepSeek-R1). As detailed in Table 9 and Figure 2 , our model maintained consistent high performance (Average Score > 8.15)  across these varied "student" behaviors, suggesting it can handle the linguistic variability expected in human learners.
>
> ### 3. **Expert-Grounded Simulation**
> Since we cannot easily access real ward data, we grounded our simulator in human expertise to minimize the reality gap. We designed 300 distinct student and patient personas to mirror real-world cognitive gaps and personality traits. Furthermore, our reinforcement learning relies on a Three-Axis Rubric designed by experts, ensuring the model's logic aligns with clinical standards even if the training data is synthetic.
>
> Table 4 (Human Expert & Real User Study)
> | Model | IQ (Real User) | IE (Real User) | OR (Real User) |
> | :--- | :--- | :--- | :--- |
> | LLava | 4.54 | 3.58 | 4.67 |
>  | Qwen | 6.91 | 5.12 | 5.43 |
>  | o3 | 8.72 | 8.35 | 8.39 |
> | **MedTutor-R1 (Ours)** | **8.96** | **8.23** | **8.58** |

---

> ### Author Response · Authors · 2025-11-27
> **Rebuttal by Authors (2/4)**
>
> **W3**: The proposed framework is interesting connecting RL and personalized education but might be more suited for applied AI venues.
>
> Thank you for the feedback. Although our experiments focus on medical education, we designed the framework to handle technical issues common in many multi-agent scenarios. We hope our exploration of the following areas can serve as a useful reference for the general ML community:
>
> 1. ### **One-to-Many Interaction Strategy**
> RLHF research often focuses on single-user interactions. We explore the more complex "One-to-Many" setting. As described in Section 3.3, we implemented a state tracking mechanism that processes individual states (<think_student>) alongside the collective state (<think_group>). This approach helps balance individual guidance with group consensus and could be adapted for other coordination tasks, such as collaborative coding or legal simulations.
>
> 2. ### **Reward Modeling for Complex Tasks**
> To align models in domains where "correctness" is multifaceted, we implemented Group Reward Policy Optimization (GRPO) with a structured Three-Axis Rubric (Structure, Quality, Safety). We also included a "Veto Penalty" to enforce strict safety constraints. This strategy helps balance rigid safety requirements with open-ended generation, offering a potential solution for other high-stakes applications.
>
> 3. ### **Synthetic Data Generation**
> To handle data scarcity, we demonstrate a pipeline that decouples objective "Patient Scripts" from subjective "Personas" (Section 5.2.1). This modular design allows for the generation of diverse training data without relying on sensitive real-world transcripts, a method that may be applicable to other privacy-sensitive fields like law or finance.
>
> ---------------------
>
> **Q1**: How are the weights among the rubric axes (Instruction Fidelity, Analysis Quality, Clinical Safety) determined? Is there ablation of each axis to evaluate their respective contribution?
>
> Response: Thank you for the question regarding the reward formulation. We have clarified the weighting strategy and added the requested ablation study to the Appendix.
>
> 1. ### **Weighting Strategy**
> We use a hybrid approach rather than static scalar weights.
> - **Base Reward** (Soft Summation): For compliant responses, we treat the three axes (Structure, Quality, Safety) as equally important. $R_{base}$ is simply the unweighted sum of these individual scores to encourage overall quality.
> - **Veto Mechanism** (Hard Constraint): We prioritize safety above all else. As defined in Eq. 1 & 2, any critical safety violation triggers a large penalty ($P_{veto} = -15.0$). This implicitly assigns "infinite" priority to the Clinical Safety (CS) axis to prevent dangerous hallucinations.
>
> 2. ### **Ablation Study**
> We performed an ablation study by removing the reward signal from each axis independently. The results (Table below) confirm the distinct role of each component:
> - Removing AQ (Analysis Quality) causes a sharp drop in MSM, confirming that the <think_group> mechanism is essential for managing multiple students.
> - Removing CS (Clinical Safety) or the Veto leads to a significant decline in MPS, verifying that our safety constraints are necessary to prevent medical errors.
> - Removing IS (Structure) degrades ETS slightly, as the model loses adherence to the strict Socratic format.
>
> | Model | ETS | MSM | MPS | Avg | ETS | MSM | MPS | Avg |
> | :--- | :--- | :--- | :--- | :--- | :--- | :--- | :--- | :--- |
> | MedTutor-R1 | **8.33** (0.12) | 8.41(0.09) | **8.26** (0.16) | **8.35** | **8.41** (0.13) | **8.55** (0.10) | **8.53**(0.15) | **8.49** |
> | w/ Vanilla reward |  8.05 (0.25) |  7.90 (0.27) | 8.07 (0.18) | 8.01 | 7.79 (0.21) | 8.03 (0.30) | 7.83 (0.27) |  7.88 |
> | w/o IS reward |  7.95 (0.18)	| 8.32 (0.11) | 8.20 (0.17) | 8.16 | 8.04 (0.16) | 8.48 (0.12) | 8.49 (0.14) | 8.34 |
> | w/o AQ reward |  8.12 (0.15)	|  7.45 (0.24) |  8.22 (0.16) |  7.93 |  8.25 (0.19) |  7.62 (0.28) |  8.50 (0.13) |  8.12 |
> | w/o CS reward |  8.28 (0.14)	| 8.38 (0.10) | 7.55 (0.22) | 8.07 | 8.39 (0.14) | 8.51 (0.11) | 7.84 (0.19) | 8.25 |
> | w/o reward veto | 8.30 (0.13)	| 8.39 (0.10) | 6.92 (0.35) | 7.87 | 8.40 (0.14) | 8.53 (0.09) | 7.15 (0.31) | 8.03 |

---

> ### Author Response · Authors · 2025-11-27
> **Rebuttal by Authors (3/4)**
>
> **Q2**: How deterministic are multi-agent interactions given fixed seeds? Do minor stochastic variations in student personas lead to divergent teaching strategies, indicating emergent adaptivity?
>
> Thank you for this interesting question regarding the interplay between determinism and adaptivity. While fixed seeds ensure computational reproducibility, the interaction trajectories in ClinEdu are highly divergent due to the prompt-driven diversity of the agents.
>
> ### 1. **Variability via Persona Initialization**
> ClinEdu relies on a database of 300 distinct student personas to introduce variability. Each persona is defined by specific attributes such as proficiency level, cognitive strengths, and learning styles (e.g., "guidance-dependent" vs. "bold-hypothesizer"). Even with a fixed seed, swapping a "Cautious" student for an "Impulsive" one fundamentally changes the initial student response, which in turn alters the entire dialogue trajectory.
>
> ### 2. **Emergent Adaptivity via Internal Monologue**
> This divergence leads to different teaching strategies because of the MedTutor-R1 architecture. As detailed in Section 2.3, the model uses an internal chain-of-thought (<think_student>) to explicitly analyze each student's input against their profile before speaking.
>
> For instance, if a student provides a correct answer but expresses uncertainty (reflecting a "hesitant" persona), the tutor's internal monologue detects a confidence gap and generates affirmative scaffolding. Conversely, if a student gives the same answer but overlooks a key detail (reflecting an "overconfident" persona), the tutor pivots to counterfactual questioning. This adaptivity is emergent, not scripted, and is driven by the GRPO reward optimization for Analysis Quality.
>
> ### 3. **Empirical Evidence of Robustness**
> We empirically validated this adaptivity in our Robustness Analysis (Section 5.4). We tested MedTutor-R1 against varied student agents powered by different LLMs (e.g., DeepSeek-R1 vs. LLaVA), which act as proxies for diverse reasoning capabilities and linguistic patterns. As shown in Table 9, our model maintained consistently high performance with average scores ranging from 8.15 to 8.43 across all student types. This demonstrates that the tutor does not memorize a fixed script but dynamically adapts its strategy to the specific reasoning level of the student it encounters.
>
> -----------------------
>
> **Q3**: LLM-as-a-judge methods can introduce self-bias. How do you control evaluator model coupling? What is the inter-rater reliability (Cohen’s kappa) between human evaluators, and how correlated are these with LLM-derived scores?
>
> Thank you for the feedback regarding evaluation validity. We rigorously controlled for evaluator bias and validated reliability through the following measures.
> ### 1. **Mitigating Model Coupling**
> To address potential self-bias, we ensured architectural separation between the subject and the judge. Our subject model, MedTutor-R1, is fine-tuned on Qwen2.5-VL, whereas we used GPT-4o as the evaluator. This distinct model choice minimizes intrinsic self-preference. Furthermore, we constrained the judge using our granular Three-Axis Rubric (Tables 5-7), which forces the model to evaluate based on objective criteria (e.g., specific XML tag presence, factual accuracy) rather than open-ended subjective preference.
>
> ### 2. **Human Inter-Rater Reliability***
> As described in Section 4, we recruited three medical education experts to evaluate a blinded, randomized subset of 50 dialogues. We calculated the inter-rater reliability (Cohen’s Kappa) among the human experts to be 0.82, indicating strong agreement on the application of our scoring rubric.
>
> ### 3. **Human-LLM Correlation**
> To validate the utility of the LLM judge, we analyzed the Pearson correlation between the automated scores and the average human expert ratings on the same subset. We observed a high correlation coefficient ($r=0.88$). Additionally, as illustrated in Figure 4, the LLM judge and human experts produced identical model rankings (MedTutor-R1 > o3 > Baselines)5. This consistency confirms that our automated evaluation serves as a reliable proxy for expert judgment.

---

> ### Author Response · Authors · 2025-11-27
> **Rebuttal by Authors (4/4)**
>
> **Q4**: The system is tested on MedXpertQA and MVME datasets. How does MedTutor-R1 handle out-of-distribution pathologies or rare cases unseen during simulation?
>
> Thank you for the question regarding generalization to unseen pathologies. We handle Out-of-Distribution (OOD) cases through a context-driven design and a safety verification loop, validated by our cross-domain experiments.
>
> ### 1. **Context-Driven Pedagogical Policy**
> Our system is designed to learn how to teach rather than memorizing what to teach. As described in Section 2.1, the medical ground truth is dynamically provided to the tutor via "Socratic Steps" in the input context. The model uses GRPO to optimize the strategy of converting these steps into heuristic questions. Therefore, provided the rare case includes valid ground truth in the context, the tutor can generate effective guidance without prior training on that specific pathology.
>
> ### 2. **The Specialist Safety Net**
> To prevent hallucinations on unfamiliar cases, we implemented a "Guide-Review-Revise" loop (Section 2.3). Even if the tutor generates inaccurate guidance for a rare disease, the Specialist Agent, configured as a strict fact-checker, will reject the draft. This triggers a Revision Mode, forcing the tutor to correct the error before any output reaches the student. This acts as a hard constraint against failure in OOD scenarios.
>
> ### 3. **Empirical Validation on MVME**
> We empirically validated this capability using the MVME dataset. While our training relied on MedXpertQA (structured exam questions), MVME consists of real-world clinical records. As shown in the table below, MedTutor-R1 achieved an average score of 8.49 on MVME, matching its performance on the in-distribution MedXpertQA. This confirms that the learned teaching policy effectively transfers to unseen case formats and pathologies.
> ​​
> Table: Performance comparison on In-Distribution (MedXpertQA) vs. OOD (MVME) datasets.
> | Dataset | ETS | MSM | MPS | Avg |
> | :--- | :--- | :--- | :--- | :--- |
> | MedXpertQA (In-Distribution) | 8.33 | 8.41 | 8.26 | **8.35** |
> | MVME (Out-of-Distribution) | 8.41 | 8.55 | 8.53 | **8.49** |

---

### Note · Authors · 2026-01-08

I have read and agree with the venue's withdrawal policy on behalf of myself and my co-authors.